# Lifted Weighted Mini-Bucket

**Nicholas Gallo**
University of California Irvine
Irvine, CA 92637-3435
ngallo1@uci.edu

**Alexander Ihler**
University of California Irvine
Irvine, CA 92637-3435
ihler@ics.uci.edu

## Abstract

Many graphical models, such as Markov Logic Networks (MLNs) with evidence, possess highly symmetric substructures but no exact symmetries. Unfortunately, there are few principled methods that exploit these symmetric substructures to perform efficient approximate inference. In this paper, we present a lifted variant of the Weighted Mini-Bucket elimination algorithm which provides a principled way to (i) exploit the highly symmetric substructure of MLN models, and (ii) incorporate high-order inference terms which are necessary for high quality approximate inference. Our method has significant control over the accuracy-time trade-off of the approximation, allowing us to generate any-time approximations. Experimental results demonstrate the utility of this class of approximations, especially in models with strong repulsive potentials.

## 1   Introduction

Many applications require computing likelihoods and marginal probabilities over a distribution defined by a graphical model, tasks which are intractable in general [24]. This has motivated the development of approximate inference techniques with controlled computational cost. Inference in these settings often involves reasoning over a set of regions (subsets of variables), with larger regions providing higher accuracy at a higher cost. This paper utilizes the Weighted Mini-Bucket (WMB) [10] algorithm which employs a simple heuristic method of region selection that mimics a variable elimination procedure.

Recently, there has been interest in modeling large problems with repeated potentials and structure, often described with a Markov Logic Network (MLN) language [16]. Such models arise in many settings such as social network analysis (e.g. estimating voting habits), collective classification (e.g. classifying text in connected web-pages), and many others [16]. In these settings, *lifted* inference refers to a broad class of techniques, both exact [3, 15, 21] and approximate [4, 11, 13, 5, 12, 9], that exploit model symmetries. Most of these methods work well when the model possesses well-defined symmetries [4, 11, 13, 5, 12, 14], but break down in unpredictable ways in the presence of unstructured model perturbations present in most practical settings. The problem of asymmetries in the approximate inference structure is compounded when higher order inference terms (for which lifted inference requires higher order model symmetry) are incorporated [14, 20, 5].

Methods to control computational cost in the presence of asymmetries are largely heuristic, such as [19] which presents a belief propagation procedure that approximates messages in a symmetric form. Other works create an over-symmetric approximate model [23, 22] on which inference is run, but provide no guarantees on its relation to the original problem. Similar to our work, [17] employ (non-weighted) mini-bucket inference; however, they too rely on over-symmetric model approximation heuristics to control computational cost.

This paper addresses the shortcomings of the methods described above with a lifted variant of Weighted mini-bucket (LWMB) that is able to (i) trade-off inference cost with accuracy in a controlled

manner in the presence of asymmetries, and (ii) incorporate higher-order approximate inference terms, which are crucial for high quality inference. This work can be seen as a high-order (property (ii)) extension of [9] (which is qualitatively identical in property (i)). Additionally, this work employs efficient region selection and representation for MLN models, and hence never grounds the graph as many others are required to do for symmetry detection (e.g. [9, 20, 14]).

## 2 Background

A Markov random field (MRF) over $n$ discrete random variables (RVs) $X = [X_1 \ldots X_n]$ taking values $x = [x_1 \ldots x_n] \in (\mathcal{X}^1 \times \ldots \times \mathcal{X}^n)$ has probability density function

$$p(X = x) = \frac{1}{Z} \prod_{\alpha \in \mathcal{I}} f_\alpha(x_\alpha); \qquad Z = \sum_{x_n} \cdots \sum_{x_1} \prod_{\alpha \in \mathcal{I}} f_\alpha(x_\alpha)$$

where $\mathcal{I}$ indexes subsets of variables and each $\alpha \in \mathcal{I}$ is associated with potential table $f_\alpha$. The *partition function* $Z$ normalizes the distribution. Calculating $Z$ is a central problem in many learning and inference tasks, but exact evaluation of the summation is exponential in $n$, and hence intractable.

### 2.1 Bucket and Mini-Bucket Elimination

Bucket Elimination (BE) [6] is an exact inference algorithm that directly eliminates RVs along a sequence $\boldsymbol{o}$ called the *elimination order*. Without loss of generality, we assume that each factor index $\alpha \in \mathcal{I}$ is ordered according to $\boldsymbol{o}$. BE operates by performing the summation (2) along each RV in sequence. The computation is organized with a set of buckets $B_1 \ldots B_n$ where initially each $B_v = \{f_\alpha \mid \alpha_1 = v\}$ is the set of model factors whose earliest eliminated RV index is $v$. Proceeding sequentially along $\boldsymbol{o}$, we multiply the factors in $B_v$, then sum over $x_v$ producing a message

$$m_{v \to w}(x_{pa(v)}) = \sum_{x_v} \prod_{\alpha \in B_v} f'_\alpha(x_\alpha) \tag{1}$$

where $pa(v)$ is the arguments of factors in $B_v$ not including $v$. The message is then placed in bucket $B_w$. If $w = \emptyset$ the message is a scalar. All such messages are multiplied to form $Z$. The computational cost is exponential in the scope of the largest message, which is prohibitive in most applications.

**Mini-Bucket Elimination (MBE)** Mini-Bucket Elimination [7] avoids the complexity of BE by upper (or lower) bounding the message (1) as the product of terms each over a controlled number $iBound$ of RVs. During elimination, factors in bucket $B_v$ are grouped into partitions $\mathcal{Q}_v = \{q_v^1 \ldots q_v^k\}$, where each $q_v^j \in \mathcal{Q}_v$ is called a *mini-bucket* and is associated with factors that (collectively) use at most $iBound + 1$ RVs. The true message is bounded using the inequality

$$\sum_{x_v} \prod_{\alpha \in B_v} f'_\alpha(x_\alpha) \ \leq \ \sum_{x_v} \prod_{\alpha \in q_v^1} f'_\alpha(x_\alpha) \ \cdot \ \prod_{j=2}^{|\mathcal{Q}_v|} \max_{x_v} \prod_{\alpha \in q_v^j} f'_\alpha(x_\alpha) \tag{2}$$

Each message is an upper bound on the exact message, hence the full procedure yields an upper bound on $Z$.

### 2.2 Weighted mini-bucket elimination (WMB)

WMB [10] generalizes MBE by using a tighter bound based on Holder's inequality

$$\sum_x g(x) \cdot h(x) \ \leq \ \sum_x^w g(x) \cdot \sum_x^{1-w} h(x), \quad \text{where} \quad \sum_x^w f(x) = \Big[ \sum_x f(x)^{1/w} \Big]^w \tag{3}$$

is the *power-sum* operator and $w \geq 0$, $h(x) \geq 0$, $g(x) \geq 0$. The power-sum reduces to standard sum when $w = 1$ and approaches $\max_x f(x)$ as $w \to 0^+$. Thus, Holder's inequality generalizes the inequality $\sum_x g(x) \cdot h(x) \leq \sum_x g(x) \cdot \max_x h(x)$ used by mini-bucket elimination (MBE).

WMB associates a weight $w_q \geq 0$ with each mini-bucket $q \in \mathcal{Q}_v$ where $\sum_{q \in \mathcal{Q}_v} w_r = 1$ for all $v$, then forms the bound

$$\sum_{x_v} \prod_{\alpha \in B_v} f'_\alpha(x_\alpha) \ \leq \ \prod_{q \in \mathcal{Q}_v} \sum_{x_v}^{w_q} \prod_{\alpha \in q} f'_\alpha(x_\alpha). \tag{4}$$

**Variational Optimization.** The weights can be optimized to provide tighter bounds. Additionally, applying WMB to any parameterization of the distribution yields a bound on $Z$, thus it makes sense to optimize over all valid parameterizations as well. Each parameterization is obtained by shifting factor potentials between mini-buckets. That is, for each $v$, associated with each mini-bucket $q \in \mathcal{Q}_v$ is the *cost-shifting* parameter $\phi_q(x_v)$ such that

$$m_{q \to q'}(x_{q'}) = \sum_{x_v}^{w_q} f_q^\phi(x_q) \quad \text{where} \quad f_q^\phi(x_q) = \phi_q(x_v)^{-1} \prod_{\alpha \in q} f'_\alpha(x_\alpha) \quad (\forall q \in \mathcal{Q}_v) \tag{5}$$

is the *reparameterized potential* of bucket $q$. The cost-shifting terms that were divided out of each $q \in Q_v$ are multiplied into an *aggregated cost-shifting* term

$$\phi_v^0(x_v) = \prod_{q \in \mathcal{Q}_v} \phi_q(x_v) \tag{6}$$

such that $\prod_{q \in \mathcal{Q}_v} f_q(x_q) = \prod_{q \in \mathcal{Q}_v} f_q^\phi(x_q) \cdot \phi_v^0(x_v)$. We then have the following bound (rather than (4)) on the exact BE message

$$\sum_{x_v} \prod_{\alpha \in B_v} f'_\alpha(x_\alpha) \ \leq \ \left[ \sum_{x_v}^{w_v} \prod_{q \in \mathcal{Q}_v} \phi_q(x_v) \right] \prod_{q \in \mathcal{Q}_v} m_{q \to q'}(x_{q'}) \tag{7}$$

Augmented with $w_v \geq 0$, this is simply another term in the product that was bounded with Holder's inequality we require $w_v + \sum_{q \in B_v} w_q = 1$. We search for the tightest bound by performing convex optimization over $(\boldsymbol{\delta}, \boldsymbol{w})$ where $\log(\phi_q) = \delta_q$ for all $q$. Gradients can be computed and a black-box solver can be used, or a fixed point iteration [10] can be used.

## 2.3 Symmetric models and lifted inference

Many models of interest, such as MLNs, are defined by repeated potentials organized in a symmetric structure. *Lifted inference* refers to a broad class of techniques that exploit this structure for exact or approximate inference. The basic idea is to represent identical terms in the model and identical terms generated during *ground* inference implicitly with a single template.

The simplest form of symmetry used for lifted inference is based on the stable vertex coloring of a graph [1] in which two vertices of the same color have identically colored neighborhoods. In the context of lifted inference, we require a stable coloring of the ground factor graph where factor nodes of the same color are required to have the same *ordered* node neighborhood and factor potential table. Nodes of the same color behave identically during approximate inference (e.g. [18, 12]).

RV nodes of the same color are grouped together to form the index $V_i \subset \{1 \ldots n\}$ and denote $\bar{\mathcal{V}} = \{V_1 \ldots V_N\}$. Factor nodes of the same color are grouped together to form $A_j \subset \mathcal{I}$ and denote $\bar{\mathcal{I}} = \{A_1 \ldots A_M\}$. Thus, given a stable partition (coloring) we can define

**Definition 2.1.** The **lifted scope** of $A \in \bar{\mathcal{I}}$ (relative to $\bar{\mathcal{V}}$) is $\sigma^A = [V_{b_1} \ldots V_{b_k}]$ where each $\alpha \in A$ has $|\alpha| = k$ and $\alpha_i \in \sigma_i^A$ for $i = 1 \ldots k$.

This simple symmetry will help us organize higher order symmetries throughout the LWMB algorithm. Note, that the lifted scope (unlike the scope of a ground factor) may have repeated elements. This occurs, for example in a complete symmetric graph, where $N = 1$ and $\sigma^A = [V_1, V_1]$.

### 2.3.1 Markov Logic Networks

A Markov Logic Network (MLN) [16] defines a large symmetric model implicitly via a compact first order logic (FOL) language. The MLN *predicates* defines the set of RVs, and each MLN *formula* defines a set of factors with identical potential. Both are parameterized compactly by a set of *logical variables* (LVs) taking values in a finite domain (for example the set of all people $\Delta_P$).

The *predicates* of an MLN represent an attribute associated with domain elements or a relationship among domain elements. The instantiation of a predicate with a domain element is the index of a model RV. For example, the attribute predicate "Sm" (for smokes) over the domain of all people ($\Delta_P$) corresponds to the set of ground RV indices $\{Sm(y) \mid y \in \Delta_P\}$ (meaning that $Sm(Ana)$ indexes $x_{Sm(Ana)}$). An example relationship predicate "Fr" (for friends) among all pairs of people is the set of indices $\{Fr(x, y) \mid x \neq y \in \Delta_P\}$.[1]

A formula specifies a soft-logic rule applied identically to all people (or groups of people). An example relating smoking habits between friends is "$(\forall\ y \neq z \in \Delta_P)\ Fr(y, z) \wedge (Sm(y) \Leftrightarrow Sm(z)),\ \gamma$". This corresponds to the set of $\mathcal{R} = \{\ [Fr(y, z), Sm(y), Sm(z)]\ \mid\ y \neq z \in \Delta_P\}$ and where each $\alpha \in \mathcal{R}$, $f_\alpha(x_\alpha) = f_\mathcal{R}(\bar{x}_\mathcal{R})$ where $f_\mathcal{R}(\bar{x}_\mathcal{R})$ is a template with log potential taking value $\gamma$ if $Fr(y, z) \wedge (Sm(y) \Leftrightarrow Sm(z))$ is true and 0 otherwise ("Fr(y,z)" corresponds to $\bar{x}_1$ in the template).

The FOL expressions defining formulas can be arbitrary, but they often have the form of simple domain constraints on LVs, with all-diff constraints on LVs ranging over identical domain (note all-diff(y,z) is equivalent to $y \neq z$ used in Fr-Smoker formula above). The stable coloring of the factor graph groups predicate RVs together and factors associated with each formula together. Formulas of this form also possess many higher order symmetries which we exploit later.

## 3 Lifted Weighted Mini-Bucket (LWMB)

This section presents a variant of WMB that operates on *lifted factors*, each of which is a group of identical ground factors, and eliminates blocks of random variables simultaneously. A key difficulty is choosing an approximating structure that guarantees symmetric messages (which can be represented as a lifted factor) are produced and, furthermore, that allows forming high order (high iBound) symmetric inference terms. We first discuss these operations in models that possess the necessary symmetric structure, then discuss modifications that allow us to control the size of the LWMB graph in the presence of model asymmetries. Algorithm 1 summarizes the LWMB tree construction algorithm (similar to ground mini-bucket construction [7]) developed in this section.

### 3.1 LWMB in Symmetric Models

First order symmetries specified by a stable partition of variables $\bar{\mathcal{V}} = \{V_1 \ldots V_N\}$ and model factors $\bar{\mathcal{I}} = \{A_1 \ldots A_M\}$ are necessary to provide a LWMB bound. We further require that the lifted scope $\sigma^A$ not have repeated elements (we relax both of these restrictions later). The computations for LWMB will be described by their equivalence to a set of ground operations. To this end, we define

**Definition 3.1.** A **lifted factor** $F_G(x_G) = \prod_{\alpha \in G} f_G(x_\alpha)$ is the product of the template potential $f_G(\bar{x})$ applied to all sets of ground RVs indexed by elements of $G$, which range over the same domain as $\bar{x}$ used to define the template.[2]

Blocks of ground RVs indexed by $V \in \bar{\mathcal{V}}$ are eliminated simultaneously, along a *lifted elimination order $O$*. We assume the lifted scope $\sigma^A$ of all lifted factors in the input and generated during inference are ordered by $O$. The computation is organized with a set of buckets $\{B_{V_1} \ldots B_{V_N}\}$ where initially each $B_V = \{F_A \mid \sigma_1^A = V\}$ is the set of lifted model factors whose earliest eliminated lifted RV block is $V$.

**Lifted multiplication**  Having collected lifted factors in lifted buckets, an RV partition $V$ will be processed by first forming lifted mini-buckets $\bar{\mathcal{Q}}_V = \{q_V^1 \ldots q_V^k\}$ each of which groups together and multiplies lifted factors. The lifted product corresponds to a product of ground terms and may, in general, not have a lifted factor representation. This situation can cause symmetries to break

**Algorithm 1** LWMB Tree Build

---

1: **Input:** Lifted model factors $\bar{\mathcal{I}}'$, RV partition $\bar{\mathcal{V}}'$, lifted elimination order $\boldsymbol{O}'$, $iBound$
2: $B_V = \{ F_A \mid \sigma_1^A = V,\ A \in \bar{\mathcal{I}}' \}$
3: Initialize empty set of messages and empty $Q_V\ \forall V \in \mathcal{V}'$
4: **for** $(V = \text{First}(\boldsymbol{O}');\ \bar{v} \neq \emptyset;\ \bar{v} \leftarrow \text{Next}(V, \boldsymbol{O}'))$ **do**
5:     **repeat**
6:         Select $q, q' \in Q_V$ s.t. $\prod_{A \in q \cup q'} F_A(x_A)$                $\triangleright$ Lifted multiply
7:             is valid and has template size $\leq iBound + 1$
8:         $B_v \leftarrow B_v \cup (q \cup q') \setminus \{q, q'\}$         $\triangleright$ Merge MBs, delete old
9:         For all $b$, replace $M_{b \to q}$ or $M_{b \to q'}$ with $M_{b \to (q \cup q')}$.   $\triangleright$ Re-route incoming messages
10:    **until**   $q = q' = \emptyset$
11:    **for** $q \in B_v$ **do**                        $\triangleright$ Simulate mini-bucket message pass
12:         $F_C = \prod_{A \in q} F_A(x_A)$        $\triangleright$ Get $C$: ground regions associated with product
13:         Set $p$ to indices where $\sigma^C \neq \sigma_1^C$
14:         $D = \{c_p \mid c \in C\},\quad \sigma^D = \sigma_p^C$      $\triangleright$ $D$ : scope indices of ground messages
15:         Add $\{D\}$ to $B_{\sigma_1^D}$ and add message pointer $m_{q \to \{D\}}$.
16:    **end for**
17: **end for**
18: **Output** Mini-Buckets $\{Q_V \mid V \in \bar{\mathcal{V}}'\}$, and messages structure $\{m_{q \to q'} \mid \forall q \in \cup_{V \in \bar{V}'} Q_V\}$

---

in arbitrarily complex ways (e.g., during lifted variable elimination; see [15],[21]). We need to understand lifted multiplication to design LWMB bounds that avoid this situation.

To compute the lifted product $F_T(x_T) = F_R(x_R)F_S(x_S)$, we require a *symmetric join*, for given index vectors $p$ and $q$, to exist. This means there exists a $T$ where for each $t \in T$, $t_p \in R$ and $t_q \in S$, and furthermore that for each $r \in R$, $|\{t \mid t_p = r, t \in T\}| = |T|/|R|$, meaning that each $r$ participates in the same number of elements of $T$ in position $p$ (and similarly for $S$). We then have

$$F_T(x_T) = \prod_{t \in T} f_T(x_t) = \prod_{t \in T} f_R(x_{t_p})^{|R|/|T|} f_S(x_{t_q})^{|S|/|T|} \tag{8}$$

where $f_T(\bar{x}) = f_R(\bar{x}_p)f_S(\bar{x}_q)$. In the simplest case, $|R| = |S| = |T|$ and there is a one-to-one mapping, corresponding to a series of standard ground multiplications. Otherwise, it corresponds to "spreading" a ground factor across many identical (up to renaming of RVs) ground multiplications. If the symmetric join does not exist, we say the lifted multiplication is *invalid*.

Only one set of $p$ and $q$ can be valid when the lifted scopes have unique elements. The lifted scope of the multiplication (if valid) will be $\sigma^T = [\sigma^R, \sigma^S]$. Hence we set $p$ and $q$ such that $\sigma_p^T = \sigma^R$ and $\sigma_q^T = \sigma^S$. This matches the lifted factors on first order symmetries, which is a necessary but not sufficient condition for higher order symmetries.

**Symmetric join with FOL formulas** If $R$ and $S$ are represented with FOL formulas and each contains only domain constraints, the symmetric join can be performed quickly (or determine none exists). If the domain constraints between $R$ and $S$ are either disjoint or identical, we simply match the two on their LVs with identical domain.

For example, multiplying factors defined by $\{[A(x), B(x)] \mid x \in \Delta^1\}$ and $\{[A(x), C(x)] \mid x \in \Delta^1\}$ produces $\{[A(x), B(x), C(x)] \mid x \in \Delta^1\}$. As another example, $\{[A(x), B(x)] \mid x \in \Delta^1\}$ and $\{[A(x), C(z)] \mid x \in \Delta^1, z \in \Delta^3\}$ will produce $\{[A(x), B(x), C(z)] \mid x \in \Delta^1, z \in \Delta^3\}$ if $\Delta^1 \cap \Delta^3 = \emptyset$. The main algorithm in section 4 uses many symmetric lifted factors of this form.

**Lifted cost-shifting** For each $V$, each lifted mini-bucket $q \in B_V$ is associated with a weight $W_q$ and cost-shifting lifted factor $\Phi_V^q(x_V)$ with template $\phi_q(\bar{x})$. We form (analogous to 5)

$$F_Q^\Phi(x_Q) = \Phi_V^q(x_V)^{-1} \prod_{A \in q} F_A'(x_A) \tag{9}$$

where $F_A'$ represents lifted factors arising from model or a message in mini-bucket, $Q$ represents the set of ground factors associated with the product of all of mini-bucket $q$'s lifted factors.

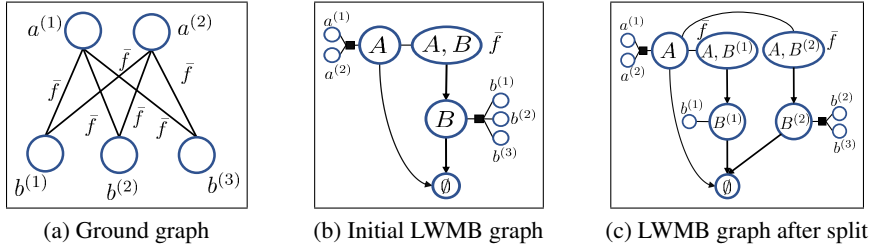

| (a) Ground graph | (b) Initial LWMB graph | (c) LWMB graph after split |

Figure 1: (a) Symmetric graph with potential $\bar{f}$ on each edge and a distinct unary potential at each node, (b) LWMB graph with partition $A = \{a^{(1)}, a^{(2)}\}$, $B = \{b^{(1)}, b^{(2)}, b^{(3)}\}$ and lifted elimination order $(A, B)$, (c) LWMB graph with $B$ partitioned into $B^{(1)} = \{b^{(1)}\}$ and $B^{(2)} = \{b^{(2)}, b^{(3)}\}$. Each RV partition is connected to its associated ground RVs with a horizontal edge through a solid square node. Each other horizontally oriented edge (e.g. between nodes $(A)$ and $(A, B)$ in panel (b)) is associated with a cost-shifting term.

**Lifted message passing.** The message $Q$ sends should be a lifted factor equal to the product of identical messages sent from each $\alpha \in Q$. The result will be a lifted factor over $D = \{\alpha \backslash \alpha_1 \mid \alpha \in Q\}$. Since each $v \in V$ appears $|Q|/|V|$ times in $Q$ (due to symmetry), each ground factor should be eliminated with weight $w_q = W_q/(|V|/|Q|)$ yielding the template

$$m'_{q \to q'}(\bar{x} \setminus \bar{x}_1) = \left( \sum_{\bar{x}_1}^{w_q} f_Q^{\Phi}(\bar{x}) \right)^{|Q|/|D|} \tag{10}$$

The power $|Q|/|D|$ arises since each $d \in D$ receives (by symmetry) $|Q|/|D|$ copies of the message. The lifted factor message, denoted $M_{q \to q'}(x_D)$ has template $m'_{q \to q'}$ applied at all indices in $D$.

## 3.2 Handling asymmetries

The exact symmetries necessary to perform lifted inference rarely exist in practice. In the extreme case, model asymmetries cause lifted algorithms to ground the model. Here, we extend LWMB to handle asymmetries (i) induced by the elimination order (to prevent, for example, grounding a complete symmetric graph), and (ii) induced by unstructured unary evidence.

**Sequential Asymmetries.** Suppose for a lifted factor $F_G$ there are $K = |\sigma^G = \sigma_1^G| > 1$ copies of the earliest variable partition in the lifted scope $\sigma^G$. In this case, any ground elimination order will treat RVs in the partition differently (hence requires grounding). The way around this is to eliminate $K$ RV's simultaneously $\sum_{x_K}^{w_q} \cdots \sum_{x_1}^{w_q} f_G(\bar{x})$, where $w_q = W_q/(|V|/|Q| \cdot K)$. This can be justified by applying Holder's inequality with any elimination order and appropriately tied weights, noting that the power-sum with tied weights commute (details omitted for space).

**Distinct Unary Evidence.** The LWMB bound can be modified to incorporate distinct single-RV potentials. The trick (similar to [9]) is to aggregate the lifted cost-shifting terms and multiply the result with the ground unary terms. That is, define

$$m_{V \to \emptyset} = \prod_{v \in V} \sum_{x_v}^{W_V} f_v(x_v) \cdot \phi_V(x_v) \tag{11}$$

where $\phi_V(\bar{x}) = \prod_{q \in Q_V} \phi_V^q(\bar{x})$.[3] Figure 1b illustrates a LWMB graph with aggregated approximate evidence terms.

## 4 Coarse to fine LWMB for MLN models

In this section we build a sequence of LWMB approximations of gradually increasing accuracy and computational cost. Starting with a LWMB tree using the coarsest possible partition, we iteratively

**Algorithm 2** Coarse To Fine LWMB for MLNs
_______________________________________________________________________________
 1: **Initialize** Choose elimination order $O$ on MLN predicates
 2: **Initialize** Build LWMB tree with MLN predicates and formulas
 3: **repeat**
 4:      Associate unary evidence with lifted RVs            ▷ To compute (11) during inference
 5:      Optimize bound over $(\bar{\boldsymbol{\delta}}, \boldsymbol{W})$
 6:      $F_Q \leftarrow F_Q^{\Phi}/(M_{q \rightarrow q'}) \; \forall q$                    ▷ See "Maintaining Monotonicity"
 7:      Set domain partition $\Delta_d \rightarrow \{\Delta_d^1, \Delta_d^2\}$       ▷ via gradient cluster or another method
 8:      Split $(\bar{\boldsymbol{\delta}}, \boldsymbol{W})$, $\bar{\mathcal{V}}$, and lifted factors that use domain $\Delta_d$       ▷ Section 4.1
 9:      Update lifted elimination order $O$
10:      Build LWMB tree with new lifted MB regions and RVs as input
11: **until** Exact answer computed
_______________________________________________________________________________

improve the approximation with *Splitting* and *Joining* operations. *Splitting* partitions the cost-shifting parameters (eq. (11)) into finer groupings, allowing a more flexible interaction with evidence. *Joining* incorporates high-order inference terms by performing LWMB Tree Build with high iBound. This procedure is summarized in Algorithm 2 and described in this section. An example of the effect of splitting on the LWMB graph in Figure 1b is show in Figure 1c.

### 4.1 Splitting

LWMB splitting is similar to the splitting operation presented in [9] for factor-graph models, but operates by partitioning a group of MLN domain elements rather than a group of variational parameters (as in [9]). That is, we partition a single domain $\Delta \in \bar{\boldsymbol{\Delta}}$ into two disjoint domains ($\Delta^{(1)} \cup \Delta^{(2)} = \Delta$ and $\Delta^{(1)} \cap \Delta^{(2)} = \emptyset$). Then, we split all lifted factors and RV partitions that use $M \geq 1$ LVs with domain $\Delta$ into $2^M$ finer lifted factors. An important property of this splitting scheme is that lifted factors with FOL form are split into lifted factors with FOL form.

**Example 1.** A variable partition $V = \{Sm(y) \mid y \in \Delta\}$ splits into $\{\{Sm(y) \mid y \in \Delta^{(i)}\} \mid i \in \{1,2\}\}$. A lifted factor $F_R$ with $R = \{[Fr(y,z), Sm(y), Sm(z)] \mid \forall y \neq z \in \Delta\}$ splits into 4 lifted factors $F_{R'}$ where $f_{R'} = f_R$, $w_{R'} = w_R$ and where $R' \in \{\{[Fr(x,y), Sm(x), Sm(y)] \mid \forall x \neq y, x \in \Delta^{(i)}, y \in \Delta^{(j)}\} \mid (i,j) \in \{1,2\}^2\}$.

Another important property is that lifted factors with $M > 1$ LVs with the same domain split into lifted factors with LVs of all distinct domains. Lifted factors of this form can participate in lifted multiplications resulting in higher order joins (section 3). For example, in Example 1 when $i \neq j$, the $x \neq y$ constraint is superfluous and can be dropped.

**Updating the lifted elimination order.** The domain split causes an RV partition $V \in \bar{\mathcal{V}}$ to split into $V'_1, \ldots, V'_{2^M}$. Since the previous LWMB bound eliminated RVs $x_V$ simultaneously, we must eliminate RVs sequentially $V'_1, \ldots, V'_{2^M}$ in the same position in $O$ relative to other RV partitions. For example, in Figure 1c $(A, B^{(1)}, B^{(2)})$ is a valid elimination order while $(B^{(2)}, A, B^{(1)})$ is not.

**Maintaining Monotonicity.** Modifications to the LWMB tree (either splitting or joining) can cause unpredictable changes in the message structure. Qualitatively, we must ensure that each new region can send a forward message, creating new regions as necessary. Such structural modifications to the flow of messages can cause an increase in the bound. To guarantee a monotonically improving bound, we replace each lifted factor with the cost-shifted mini-bucket functions divided by their forward message $F_Q \leftarrow F_Q^{\Phi}/(M_{q \rightarrow q'})$ (a similar idea was used in the ground case in [8]). This is simply a reparameterization of the model, but ensures that each node in the LWMB tree sends a uniform message of all 1's. Hence, after split or join we can simply call LWMB Tree Build with the reparameterized terms and guarantee a monotonic bound improvement. In practice, we apply this technique only to nodes in the tree affected by a split or join operation, leaving the message structure of other nodes unchanged.

**Choosing the domain partition.** The goal is to cheaply find a reasonably good split grouping. A similar problem was considered in [9]. They compute a 2-way clustering of the gradient of their inference objective with respect to variational parameters. The main idea is if parameters are constrained to be identical (for computational improvement with lifting) but their greedy unconstrained next step

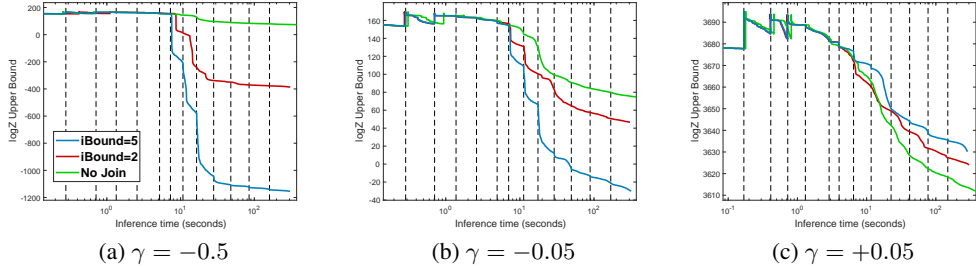

| (a) $\gamma = -0.5$ | (b) $\gamma = -0.05$ | (c) $\gamma = +0.05$ |

Figure 2: Repulsive ($\gamma < 0$) vs. attractive ($\gamma > 0$) collective classification example. (a)-(b)High iBound is extremely important in the presence of strong repulsive potentials, (c) but performs slightly worse than the baseline ("No join") in the attractive case. Dashed black lines indicate when a batch of splitting occurs for the blue curve (split transitions for other curves occur at similar locations).

would have been similar, then the lifting restriction incurs little error. Here, we perform a similar operation, but a 2-way split of a domain partition $\Delta_d$ induces a split of many parameters, associated with lifted RVs that use domain $\Delta_d$, into 2 groups. Our clustering objective is a sum over squared error of all these terms (details omitted for space).

## 5    Experiments

This section provides an empirical illustration of our LWMB algorithm. We demonstrate the superiority of utilizing high order LWMB approximations for models with repulsive potentials. In models with strictly attractive potentials, low order approximations work slightly better, likely due to their ability to split more, obtaining better approximations of the evidence.

**Setup.**    We consider a standard collective classification MLN with formula $(\forall x \neq y \in \Delta) L(x, y) \wedge (C(x) \Leftrightarrow C(y))$, $\gamma$. If $\gamma < 0$, a hard true observation on link $L(x, y)$ induces a repulsive potential between $C(x)$ and $C(y)$. $\gamma > 0$ induces attractive potentials. We run experiments with $N = |\Delta| = 512$, with clustered evidence. We randomly assign elements of $\Delta$ to one of $K = 16$ clusters. Evidence on $C$ predicate has potentials $[0; a]$. Each cluster generates a (scalar) center on $\mathcal{N}(0, 2)$ each member of the cluster is then perturbed from its center by $\mathcal{N}(0, 0.4)$ noise. Relational evidence on $L$ is generated as follows: each of the $K$ blocks has all true evidence with each other block with probability $0.25$. We then flip $25\%$ of the evidence uniformly at random.

**Optimization.**    We call a black-box convex optimization (using non-linear conjugate gradients) allowing a maximum of 1000 function evaluations. The gradient of the LWMB objective (derivation omitted for space) is computable in time roughly equal to the cost of evaluating the objective.

**Timing.**    We report only time spent doing inference (optimization), and allow each method 250 seconds total. Inference is the algorithmic bottleneck, and code has been written in C++. The rest of Algorithm 2 simply updates the LWMB structure (performing less work than a single inference iteration) but is coded in MATLAB and thus yields unreliable timing. We note that other works on lifted inference report only inference time [12, 2, 9], yet incur significant overhead of symmetry detection (that could require touching the ground model) which we never do.

**Results.**    Figure 2 shows results for (a) strongly repulsive case, (b) weakly repulsive case, (c) weakly attractive case. Strongly attractive ($\gamma = 0.5$) was qualitatively similar to (c) and omitted for space.[4] We see in (a) that lifted inference with higher order terms significantly outperforms the fully relaxed ("No Join") method. In the attractive case (c) higher order performs slightly worse. We believe this is because the cheaper inference method builds approximations of finer resolution. For panel (c), by the end, Blue performed 31 splits, Red 60 splits, and Green 73 splits.

**Acknowledgements**

This work is sponsored in part by NSF grants IIS-1526842, IIS-1254071, by the United States Air Force under Contract No. FA9453-16-C-0508, and DARPA Contract No. W911NF-18-C-0015.

## Footnotes

[1]In general, can have $> 2$ LVs and $> 1$ domain type

[2]$x_G$ abuses notation, refering to $x_{G'}$ where $G' = \cup_{\alpha \in \mathcal{G}} \alpha$ is the set of all RVs used by elements of $G$

[3]Identical potentials, such hard evidence, can be grouped into a single computation (omitted for space).

[4]Bumps in the curves are due to optimization initialization. Numerical issues arise when power-sum weights are small. Hence at the beginning of each optimization we floor them at $10^{-4}$.

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
