[Reviews · NeurIPS 2018]

Reviewer 1



This paper performs a lifted weighted mini-bucket algorithm. This is along the lines of other works that seek to lift various inference algorithms (e.g., variable/bucket elimination, belief propagation, etc) so that they operate more efficiently on relational models such as Markov Logic Networks (MLNs). One distinction here is that they propose an anytime algorithm that starts with the coarsest lifted weighted mini-bucket approximation, and then incrementally tightens the bound by refining the operation, by trading off computation. The paper appears to be thorough in its presentation, and considers the different aspects of lifting the weighted mini-bucket algorithm. Lifted infernece is quite relevant to the field of probabilistic graphical models, and this paper appears to make an advance in this regard.

Reviewer 2



The paper proposes a lifted version of the weighted mini-bucket inference algorithm. Weighted mini-bucket is a variant of Variable Elimination that can trade off computational cost (e.g., achieving runtime sub-exponential in the treewidth) for accuracy. It essentially uses Holder inequality and variational approximations to represent "messages" that do not fit in the available memory budget. The main idea is to extend the approach to relational models (e.g., Markov Logic Networks) to take advantage of the additional structure, specifically, the fact that ground factors are produced from first-order templates and therefore share significant structure. The work appears to be solid, but it is difficult to parse. The notation used is fairly heavy (although that's probably unavoidable), and the paper doesn't provide enough examples to clarify some of the concepts, e.g., what the LWMB tree would look like in some simple cases. Concepts like "monotonicity" are not really explained/defined. The paper might be understandable to the lifted inference sub-community, but for a non-expert, it's quite difficult to follow Sections 3 and 4 (the other parts are actually well written). The fact that other implementation details are omitted ("In practice, we try to keep as much of the structure in tact as possible (details omitted), applying this reparameterization technique parsimoniously."), doesn't help. The experimental results appear to be weak. The key question that is not answered is, does this advance the state of the art in some way? It is difficult to assess without a comparison with other existing (lifted) inference methods. Comparing with standard variational inference algorithms would also provide some perspective on the performance of the proposed method (including lower bounds in figure 1 to get a sense of how big the improvements are). Finally, the experimental evaluation is conducted on synthetic models only. It's therefore unclear how well the method would perform in practice, and how much the structure exploited by the proposed technique is useful on real world problems.

Reviewer 3



The paper presents a lifted version of the weighted mini-bucket elimination algorithm. Exploiting symmetries within probabilistic inference is important. In particular, with respect to the present paper, the standard approach for inference in probabilistic formalisms with first-order constructs is lifted variable elimination (LVE) for single queries. To handle multiple queries efficiently, the lifted junction tree algorithm (LJT) employs a first-order cluster representation of a model and LVE as a subroutine. The present paper extends this family of approach towards a lifted variant of weighted mini-bucket. That is, it shows how to employ its tighter bounds for probabilistic inference. This is interesting. And yes, the paper mentions major related work. However, it doe not discuss some important related work at a sufficient level of details. Consider e.g. [5, 13] but also Prithviraj Sen, Amol Deshpande, Lise Getoor: Bisimulation-based Approximate Lifted Inference. UAI 2009: 496-505 In terms of the bound used, as it is closely related to counting numbers, the authors should clarify the connection to [13] and related approaches in more details. This line of research has also shown that symmetries can be exploited for concave energies and for variational optimization of them. This discussion is missing. Also, the authors should mention Seyed Mehran Kazemi, David Poole: Knowledge Compilation for Lifted Probabilistic Inference: Compiling to a Low-Level Language. KR 2016: 561-564 David B. Smith, Parag Singla, Vibhav Gogate: Lifted Region-Based Belief Propagation. CoRR abs/1606.09637 (2016) As far as I see it, the underlying theory presented, next to the join operations, are very similar if not identical. The authors should develop where (on the underlying algebraic level) the differences are. As far as I see it, the present paper introduces a novel approach to compute counting numbers, which could then be used e.g. in Theorem 9 of [13]. This is also related to the asymmetric case presented. The sequential case is interesting but not discuss in detail. The unary evidence case is very much related to the domain graph presented by Bui et al. (see above) as well as by Martin Mladenov, Kristian Kersting, Amir Globerson: Efficient Lifting of MAP LP Relaxations Using k-Locality. AISTATS 2014: 623-632 Overall, however, the contribution and also the particular combination appears to be very interesting. The experimental evaluation falls a little bit too short as no real world data has been considered. Would be great to add something evaluations here. Nevertheless, overall, the paper makes a nice progress for lifted inference. The main downside is the weak discussion of related work, which makes it hard to see the novel contributions. This is also reflected in the fact that now summary of the results are presented. The experimental evaluation is too short for the claims raised in the intro, but still kind of fine. I am confident that this will work on other models, too. Furthermore, there are forthcoming IJCAI 2018 papers such as Tanya Braun, Ralf Möller Parameterised Queries and Lifted Query Answering IJCAI 2018 They introduce the idea of parameterised queries as a means to avoid groundings, applying the lifting idea to queries. Parameterised queries enable LVE and LJT to compute answers faster, while compactly representing queries and answers. Would be great if the authors can mention them, although the work was indeed done independently.